# Fidelity of Cotranslational Protein Targeting to the Endoplasmic Reticulum

**DOI:** 10.3390/ijms23010281

**Published:** 2021-12-28

**Authors:** Hao-Hsuan Hsieh, Shu-ou Shan

**Affiliations:** Division of Chemistry and Chemical Engineering, California Institute of Technology, Pasadena, CA 91125, USA; hhsieh@caltech.edu

**Keywords:** protein targeting, signal recognition particle, nascent polypeptide-associated complex, ribosome, endoplasmic reticulum, membrane proteins, fidelity

## Abstract

Fidelity of protein targeting is essential for the proper biogenesis and functioning of organelles. Unlike replication, transcription and translation processes, in which multiple mechanisms to recognize and reject noncognate substrates are established in energetic and molecular detail, the mechanisms by which cells achieve a high fidelity in protein localization remain incompletely understood. Signal recognition particle (SRP), a conserved pathway to mediate the localization of membrane and secretory proteins to the appropriate cellular membrane, provides a paradigm to understand the molecular basis of protein localization in the cell. In this chapter, we review recent progress in deciphering the molecular mechanisms and substrate selection of the mammalian SRP pathway, with an emphasis on the key role of the cotranslational chaperone NAC in preventing protein mistargeting to the ER and in ensuring the organelle specificity of protein localization.

## 1. Introduction

Over ~30% of the newly synthesized proteins in eukaryotic cells are initially delivered to the endoplasmic reticulum (ER) membrane, where they initiate their journeys through the endomembrane system including the ER, the Golgi apparatus, secretory vesicles, and the plasma membrane [1,2]. These membrane and organellar proteins are prone to misfolding, aggregation, and consequent degradation in the cytosol where their biosynthesis begins [3]. For this reason, proteins destined to the endomembrane system predominantly use a cotranslational pathway of targeting and translocation mediated by signal recognition particle (SRP), giving rise to the ribosome-studded morphology of the rough ER (Figure 1A). By coupling the synthesis of proteins to their localization, the SRP pathway minimizes the off-pathway interactions of nascent membrane and organellar proteins in the cytosol and provides the most efficient mechanism for membrane protein biogenesis, a process that is kinetically demanding and energetically costly [3,4].

How fidelity is achieved in protein localization has been a long-standing question that is challenging to address conceptually and experimentally. SRP-dependent proteins contain a transmembrane domain (TMD) on integral membrane proteins or an ER signal sequence, characterized by a contiguous stretch of hydrophobic amino acids, on secretory and organellar proteins. However, signal sequences and TMDs are divergent in length, sequence and amino acid composition [5,6]. The degenerate nature of these targeting signals demands that protein targeting machineries, such as SRP, distinguish between the correct and incorrect substrates based on minor differences in the molecular features of signal sequences. In addition, eukaryotic cells contain multiple membrane-enclosed compartments, such as mitochondria and peroxisomes, to which a nascent protein could be targeted (Figure 1A). The recent observation that SRP depletion leads to the mis-localization of proteins to mitochondria [7] provides a salient example of the promiscuity of the targeting signals and pathways. Finally, translation termination effectively abolishes the SRP pathway. In addition, it has been reported that SRP loses targeting competence after the nascent chain reaches a critical length of ~130 amino acids (aa) [8,9]. These effects impose a limited time window for SRP to complete the targeting reaction (Figure 1A). The significantly slower translation elongation rate for eukaryotic (3–6 aa/s) than bacterial (10–20 aa/s) ribosomes implies that this time window is significantly longer in eukaryotic cells, and could increase the probability of mis-targeting.

**Figure 1 ijms-23-00281-f001:**
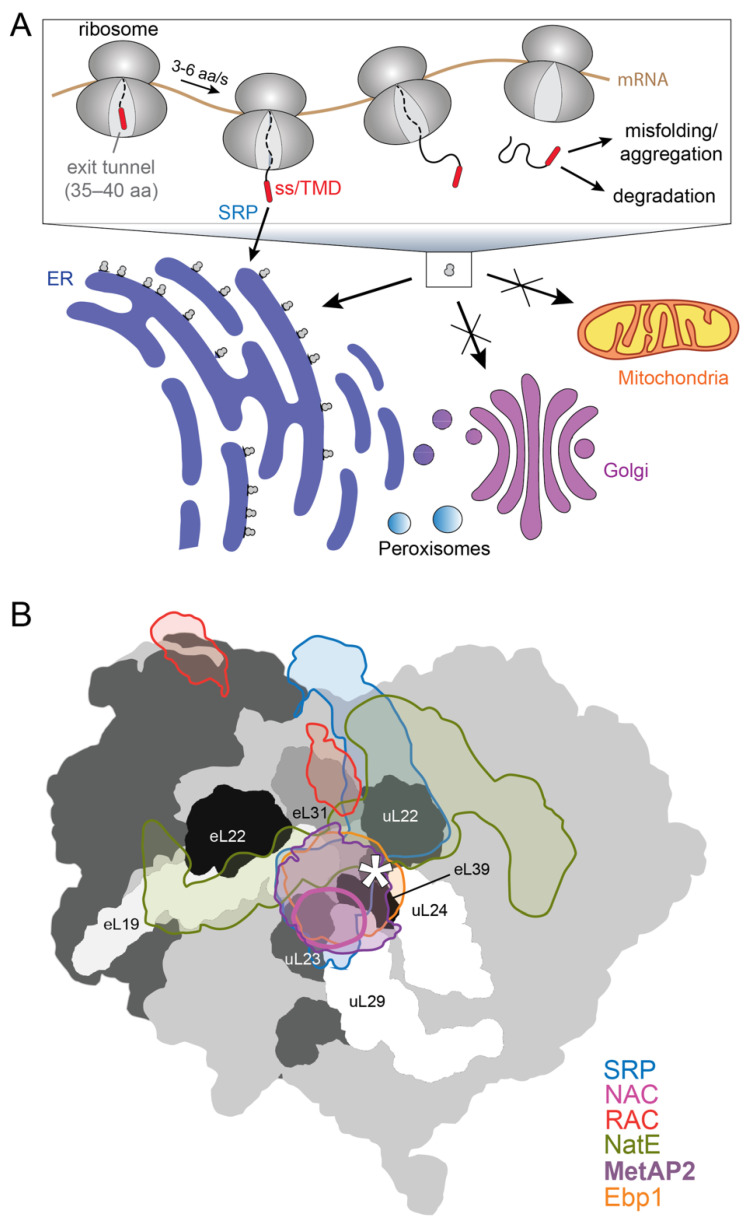
Overview of cotranslational ER targeting and ribosome-associated protein biogenesis factors (RPBs). (**A**) Overview of cotranslational protein targeting in eukaryotic cells. Proteins destined to the endomembrane system are initially targeted to the ER during translation, based on recognition of highly hydrophobic signal sequences (SS) or TMDs on the nascent chain by the SRP pathway. Cotranslational protein targeting is likely in kinetic competition with translation elongation, and failure to complete the targeting reaction within the duration of protein synthesis can lead to the misfolding, aggregation and downstream degradation of nascent secretory and membrane proteins. (**B**) Overlay of known RPB structures onto the surface facing the nascent polypeptide exit site (marked by ‘*’) of the 80S eukaryotic ribosome (PDB-4UG0; *light grey* and *dark grey* indicate the 60S and 40S subunits, respectively). Electron densities of ribosomes bound with SRP (EMD-3037), NatE (EMD-4745), NAC (EMD-4938), RAC (EMD-6105), Ebp1 (EMDB-10321), or MetAP2 (PDB-1BN5) are aligned according to the 60S density. The density of MetAP2 is derived from homology modeling with Arx1-ribosome structure (PDB-5APN). The densities of RAC and NAC contain only part of the complex due to low resolution of the EM density map. The silhouettes of the individual RPBs are shown in the indicated colors. Ribosomal proteins in the vicinity of the tunnel exit are colored in different shades of grey and indicated following the nomenclature proposed in Ban et al. [10], with ‘L’ indicating the large ribosomal subunit, ‘u’ indicating universally conserved ribosomal proteins, and ‘e’ indicating eukaryote-specific ribosomal protein subunits.

SRP also provides a salient example of an emerging concept: protein biogenesis begins early on translating ribosomes, long before synthesis of the nascent polypeptide is completed [11,12,13]. Indeed, the vicinity of the nascent polypeptide exit tunnel of the ribosome provides a platform to recruit multiple ribosome-associated protein biogenesis factors (RPBs), including cotranslational chaperones (nascent polypeptide associated complex (NAC) and ribosome-associated complex (RAC)), protein targeting and translocation machineries (SRP and Sec61p), nascent protein modification enzymes (methionine aminopeptidase (MetAP) and N-acetyl transferase E (NatE)), and quality control factors (Figure 1B) [13]. The RPBs dock at conserved and overlapping sites near the nascent polypeptide exit tunnel on the ribosome, and their engagement with the nascent chain directs the newly synthesized protein to distinct biogenesis pathways. How a nascent protein recruits the correct set of RPBs and thus commits to the proper biogenesis pathway in a timely manner is an emerging question at the heart of accurate protein biogenesis.

In this article, we review recent progress in understanding the molecular mechanism and substrate selection of the eukaryotic SRP pathway, with an emphasis on results demonstrating how regulation of SRP by the cotranslational chaperone NAC enhances the fidelity of protein targeting to the ER. Based on these and recent work on related pathways, we suggest that cells evolved multiple mechanisms to overcome the physicochemical challenges in recognizing degenerate targeting signals. These include allosteric regulation by macromolecular crowding at the ribosome exit site, kinetic competition with translation elongation, rivalry of opposing targeting pathways with overlapping but distinct substrate preferences, and surveillance and error correction mechanisms at the organelle membrane. It is likely that each individual mechanism generates a modest degree of specificity, but collectively, the combination of these mechanisms ensures the accuracy of membrane protein localization and organelle biogenesis.

## 2. SRP-Dependent Cotranslational Protein Targeting

SRP is a universally conserved ribonucleoprotein particle comprised of the 7SL SRP RNA on which six protein subunits (SRP19, SRP54, SRP68, SRP72, SRP9, SRP14) are assembled (Figure 2A). SRP is responsible for the targeted delivery of newly synthesized membrane and secretory proteins to the SecYEG translocase at the bacterial plasma membrane, or the Sec61p translocase at the eukaryotic ER membrane. The universally conserved core of SRP is a GTPase, SRP54, with two structural and functional domains: a methionine-rich M-domain, which binds the SRP RNA and provides the docking site for ER signal sequences (Figure 2A) [14,15,16,17,18,19]. The M-domain is connected via a flexible linker to a special GTPase domain, termed NG, which can interact with ribosomal protein uL23 near the exit site [20,21,22,23]. SRP54-NG assembles a stable, GTP-dependent dimer with a highly homologous NG-domain in the SRP receptor (SR; Figure 2A) [24,25,26,27,28]. The two NG-domains undergo cooperative conformational rearrangements in their heterodimer that culminates in their reciprocal GTPase activation, followed by GTP hydrolysis that drives the disassembly and recycling of SRP and SR [2,25,29,30,31,32,33,34,35,36]. Extensive work on this simplest SRP system in bacteria showed how this dimerization-activated GTPase cycle ensures the fidelity of the prokaryotic SRP pathway: ribosomes bearing an SRP-dependent signal sequence not only bind SRP more strongly, but also mediate SRP–SR assembly at rates that are 100–1000 fold faster than those on signal-less ribosomes or ribosomes with suboptimal signal sequences (Figure 3A,B, *E. coli*) [37,38]. Furthermore, ribosomes bearing an SRP-dependent substrate effectively delays GTP hydrolysis in the SRP•SR complex until the arrival of the SecYEG translocase, and thus effectively couples the SRP/SR GTPase cycle to productive protein translocation [33,37,38,39]. In contrast, SRP•SR complex assembled on signal-less ribosomes prematurely hydrolyzes GTP, aborting the targeting reactions to help reject nascent proteins that lack an ER targeting signal [37,38].

While these core GTPases in SRP and SR are highly conserved across species, both SRP and SR undergo extensive expansions in size and complexity during evolution. While bacterial SRP is a complex of the 4.5S RNA with the SRP54 homologue Ffh, eukaryotic SRP contains a larger 7SL RNA on which five additional protein subunits (SRP19, SRP68/72, SRP9/14) are assembled (Figure 2A) [40,41]. While the bacterial SRP receptor is a single protein FtsY in which the NG-domain is preceded by two amphiphilic lipid-binding helices [42,43,44,45,46], eukaryotic SR is a heterodimer of SRα and SRβ subunits (Figure 2A). SRβ is a single-pass transmembrane protein anchored at the ER. SRα binds tightly to SRβ via its N-terminal X-domain [47,48], which is connected to the NG-domain through a ~200-residue intrinsically disordered linker that contains sites for ribosome interaction and sensing [49,50]. Extensive progress has been made in elucidating the function of many of the eukaryote-specific SRP components and deciphering the molecular mechanism of the mammalian SRP pathway in recent years, owing in large part to the ability to reconstitute human SRP and SR with recombinant components [51]. This enabled detailed biochemical and biophysical analyses of the molecular events in the pathway, the identification of new targeting intermediates and the elucidation of their structures, together generating a molecular model for the pathway that incorporates structural, dynamic, and kinetic information.

To summarize, free SRP appears to be locked in a latent conformation that is inactive in its interaction with SR (Figure 2B) [51]. The particle is activated upon binding to the ribosome, on which it can sample a variety of conformations with its NG domain positioned differently relative to the proximal end of SRP (Figure 2B, step 1) [51]. The emergence of an ER signal sequence drives SRP into the ‘Proximal’ conformation, in which SRP54 NG docks at uL23 in close proximity to the ribosome exit site (step 2) [20,21,51]. In this conformation, SRP initiates assembly with SR via the interaction between their NG domains (step 3). Early SRP–SR association is assisted by a molecular recognition feature (MoRF) in the SR linker, which contacts both the M- and NG-domains of SRP54 to stabilize the earliest stage of targeting (Figure 2B, ‘Early’) [49,52]. Formation of a stable NG dimer drives a series of conformational rearrangements, leading to the detachment of the NG-dimer from the ribosome exit site and its docking onto the membrane-proximal X and β-domains of SR, resulting in a global compaction of the SR (Figure 2B, ‘Compact’) [52]. A new molecular surface is generated in the resulting NG•Xβ complex, allowing it to dock onto the distal end of SRP where SRP68/72 is located (Figure 2B, step 5) [21,52,53]. In this ‘pre-handover’ conformation of the targeting complex, the ribosome is brought close to the membrane surface, and the ribosome exit site is vacated and thus primed to initiate interaction with the Sec61p translocation machinery (step 6).

Thus, eukaryotic cotranslational protein targeting requires multiple largescale conformational rearrangements in both SRP and SR, which allow this targeting machine to transition successively through the cargo recognition, targeting, and cargo handover stages in the targeting cycle. These structural and functional transitions are driven by the dimerization-activated GTPase cycle of SRP/SR, the translating ribosome, and possibly other components of the pathway. Notably, multiple mutations in SRP54 NG are linked to severe syndromic neutropenia with Shwachman–Diamond-like features; these mutations block either the assembly of the NG heterodimer [54,55] or the conformational rearrangements that lead to the pre-handover complex (Figure 2B, ‘⊥’) [52], demonstrating the critical role of the GTPase-driven conformational rearrangements in the proper functioning of SRP. As described in the next section, these conformational rearrangements also provide multiple opportunities for allosteric regulation of this targeting machine, for example by additional RPBs at the ribosome exit site.

The ability to quantitatively measure the individual molecular events in the mammalian SRP pathway also enabled a comparison of the molecular interactions of the mammalian and bacterial SRP, which raised intriguing questions as to how high fidelity is achieved during cotranslational protein targeting (Figure 3A,B). As described earlier, bacterial SRP and SR form a self-sufficient system that can generate a high level of targeting specificity by using a combination of differential binding, induced fit, and kinetic proofreading mechanisms (Figure 3A,B, *E. coli*). In contrast, ribosomes with and without an ER targeting signal differ only ~four-fold in the binding of human SRP, and ~two-fold in activating the assembly between SRP and SR (Figure 3A,B, human) [56,57]. These results suggest that, unexpectedly, mammalian SRP and SR by themselves are insufficient to generate the specificity required for high fidelity protein targeting to the ER. As described in the next Section, SRP requires a cotranslational chaperone, the nascent polypeptide associated complex (NAC), to act as a triage factor during substrate selection in eukaryotic cells.

**Figure 3 ijms-23-00281-f003:**
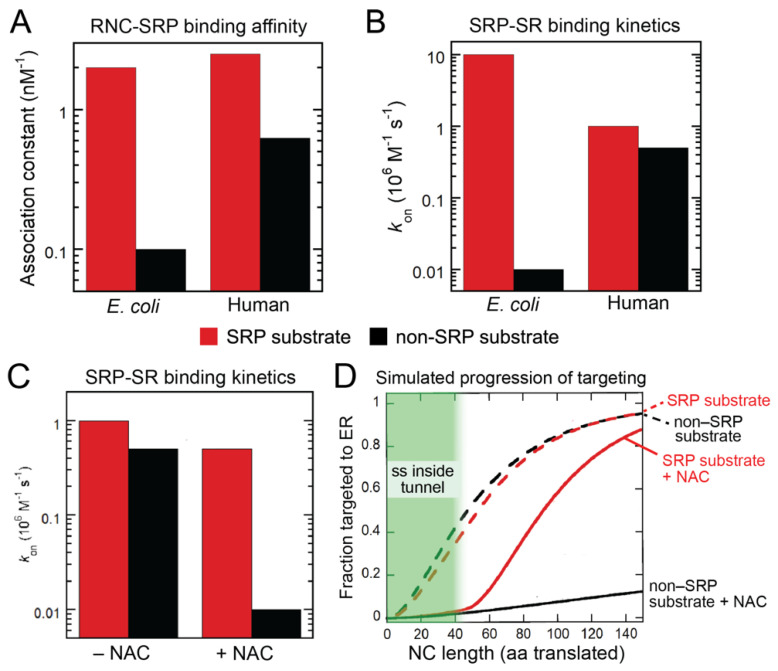
NAC regulates the activity of mammalian SRP to help reject signal-less ribosomes from ER targeting. Comparison of the specificity during the cargo recognition (**A**) and targeting steps (**B**) between the *E. coli* and human SRPs. The rate and equilibrium constants were measured with purified RNC, SRP and SR in vitro and show the promiscuity of human SRP in both steps. Adapted with modifications from the data in [56,57]. (**C**) NAC enhances the specificity of human SRP during the SR recruitment step from <2-fold to ~50-fold. Adapted from [56]. (**D**) Simulation of the progression of cotranslational protein targeting to the ER with and without NAC present. The simulation is based on a kinetic model of SRP-dependent protein targeting using experimentally determined rate and equilibrium constants of the cargo binding and SRP–SR assembly steps, followed by a commitment step in which the RNC-SRP–SR complex loads the translating ribosome onto Sec61p. The enhanced selectivity of ER targeting in the presence of NAC arise from two effects: suppression of pre-mature targeting before a signal sequence emerges from the ribosome, and inhibition of non-specific SRP–SR association on ribosomes that do not expose an ER targeting signal. Adapted from [56].

## 3. NAC: A Triage Factor during Cotranslational Protein Targeting

NAC is an abundant cotranslational chaperone expressed at equimolar concentrations relative to the ribosome in eukaryotic organisms [11,58]. Given its abundance and high ribosome binding affinity (*K_d_* ~1 nM; [56]), NAC can bind to virtually every ribosome in eukaryotic cells. How NAC interacts with the ribosome is incompletely understood. NAC is a heterodimer of α and β subunits, both containing a central NAC domain that dimerizes into a β-barrel structure (Figure 4A, pink and magenta) [59]. In addition, NACβ contains an N-terminal extension harboring a conserved basic motif crucial for its ribosome binding (Figure 4A, ‘++’) [60]. Crosslinking data suggested multiple interaction sites of NAC on the ribosome, including uL23, uL29, eL31 and eL39, all of which are located near the exit tunnel but on opposite sides [61]. A recent cryoEM structure revealed an unexpected mode of NAC interaction: the N-terminal tail of NACβ inserts deeply into the nascent polypeptide exit tunnel of the 60S ribosomal subunit [62]. In support of the structural observation, the N-terminus of NACβ can crosslink to nascent chains on ribosome-nascent chain complexes (RNCs) as short as 10 amino acids (Figure 4B, (1) and (2)), suggesting that NAC acts at the earliest stage of protein synthesis [62]. The crosslink to the NACβ N-terminus becomes weaker when the nascent chain exceeds 30 amino acids in length, suggesting that the inserted tail of NACβ is pushed out of the exit tunnel during translation elongation, and that NAC switches to a distinct mode(s) of interaction once the nascent chain emerges from the tunnel exit (Figure 4B, (3) and (4)).

NAC is generally described as a chaperone-like molecule that assists in the maturation of newly synthesized proteins [63,64]. The embryonic lethality of NAC mutants in *C. elegans*, *Drosphila melanogaster* and mice demonstrates an essential function of this chaperone in higher eukaryotic organisms [65,66,67,68]. However, the precise cellular roles and biochemical activities of NAC still await to be clearly defined. A variety of functions have been ascribed to NAC, including the de novo folding of nascent proteins and the biogenesis/maturation of ribosomes [69,70]. NAC weakly interacts with a variety of proteins and helps maintain the solubility of aggregation-prone proteins, such as alpha-synuclein and polyQ [71,72]. NAC is also proposed to be a proteostasis sensor that relocalizes from the ribosome to aggregated puncta in the presence of proteostasis stress [73]. These observations potentially reflect a small heat shock protein-like activity of NAC off the ribosome. Other suggested roles for NAC include protein import to mitochondria [74,75,76,77], NACα as a transcription activator [78,79], and suppression of apoptosis [67]. Many of these proposed roles, including the direct or indirect involvement of NAC in these processes, remain to be explored.

The best studied function of NAC is its role in the regulation of protein targeting to the ER. NAC was initially identified in rabbit reticulocyte lysate as a factor that prevents the nonspecific engagement of SRP with nascent chains that lack an ER signal sequence, and whose depletion leads to the mistargeting of cytosolic and mitochondrial proteins to ER microsomes [80]. Many ensuing studies corroborated the involvement of NAC in regulating protein sorting to the ER [81,82,83,84,85,86,87,88] and further showed that NAC binds to ribosomes with short nascent chains and forms a protective environment for regions of the nascent polypeptide just emerging from the tunnel exit [58,89]. Significantly, knockdown of NAC in *C. elegans* led to ER stress and the mislocalization of reporter proteins with a mitochondrial signal sequence to the ER [65,90], providing strong support for the role of NAC as a specificity factor during ER targeting.

Despite these earlier works, the mechanism by which NAC prevents protein mistargeting remained controversial. Nevertheless, a globular domain of NAC is located at the ribosome tunnel exit in its ‘inserted’ conformation described above (Figure 4B, (1)), in a position that can block the binding of SRP or Sec61p to the ribosome [62]. This and the observed antagonistic effect of NAC on the binding of signal-less ribosomes to SRP and to the ER membrane [81,82,83,84,85,86,87,88] gave rise to a primarily competitive model, in which NAC excludes SRP and Sec61p from binding to ribosomes without an ER signal sequence. However, recent quantitative measurements suggested otherwise. While NAC weakened the binding affinity of SRP for RNCs, in agreement with earlier observations, the binding antagonism saturated at 4–6 fold; this saturation behavior is in contrast to expectations from a strictly competitive model, which predicts that the observed binding affinity will continue to decrease with increasing concentrations of the competitor [56]. In addition, the effects of NAC on SRP binding affinity were similar between ribosomes with and without an ER signal sequence and insufficient to explain the ability of NAC to specifically suppress the targeting of signal-less nascent chains [56]. Finally, co-binding of SRP and NAC on the same RNC can be observed in single-molecule colocalization experiments, and efficient FRET was also observed between the two factors on the RNC [56], indicating that they are positioned in close proximity to each other on the same ribosome.

These observations indicate that NAC does not act solely by excluding SRP from ribosome binding but instead, exerts regulation via an allosteric mechanism. Indeed, under conditions where SRP and NAC are co-bound on the ribosome, NAC specifically reduced the kinetics of SRP–SR assembly on ribosomes without an ER signal sequence, increasing the discrimination against signal-less ribosomes to ~50-fold in this membrane-targeting step (Figure 3C; [56]). In addition, significant promiscuous SRP–SR association were observed on ribosomes with nascent chains shorter than 35 amino acids, when the targeting signal is still buried inside the nascent polypeptide exit tunnel (Figure 3D, area shaded in green) [56]. NAC also strongly suppresses these pre-mature targeting events and thus delays the onset of targeting [56]. Kinetic modeling of the SRP pathway, based on these experimentally measured parameters, showed that the combination of these regulatory effects of NAC are necessary and sufficient to generate a high degree of specificity during cotranslational protein targeting to the ER (Figure 3D), whereas in the absence of NAC, both ribosomes with and without an ER signal sequence are delivered to the ER within ~100 amino acids of their synthesis (Figure 3D, dashed lines) [56].

The allosteric regulation of SRP by NAC was directly probed in single molecule FRET measurements (Figure 5A) [56]. A pair of FRET dyes, engineered between SRP54-NG and SRP19, was used to specifically monitor the formation of the ‘Proximal’ conformation of SRP that is most active for SR recruitment [51]. On ribosomes exposing an ER targeting signal, SRP is dominated by the high FRET population corresponding to the Proximal conformation, and its conformational distribution is not substantially affected by NAC (Figure 5A, right panel). On ribosomes exposing a mutated signal sequence, in contrast, SRP is conformationally much more dynamic and heterogeneous, sampling low-, medium-, and high-FRET states all with substantial frequency [51,56]. Significantly, ~30% of SRP samples in the Proximal conformation were conducive to SR binding even when bound to signal-less ribosomes (Figure 5A), which may explain the low substrate specificity during SRP–SR assembly. However, NAC largely eliminated the population of SRP that resides in the Proximal conformation on signal-less ribosomes, forcing SRP into low- and medium-FRET states that are presumably inactive in binding with SR. (Figure 5A) [56].

Collectively, the recent results provide strong evidence that NAC acts as a triage factor that enforces the correct timing and specificity of SRP-dependent protein targeting (Figure 5B). In addition to the binding antagonism proposed previously, NAC further exerts its regulation allosterically, by remodeling the conformational landscape of SRP on the ribosome and preventing SRP from adopting the targeting-active conformation in the absence of an exposed ER signal sequence (Figure 5B). This ensures that SRP is activated to initiate targeting only upon the emergence of a correct signal sequence from the ribosome exit tunnel, thus preventing the promiscuous ER localization of ribosomes translating cytosolic proteins, or proteins destined to other organelles such as mitochondria (Figure 5B).

## 4. Perspectives and Open Questions

Emerging data show that the observation with NAC is not an isolated example, but rather, represents a general mechanism whereby the fidelity of individual protein biogenesis pathways can be reshaped by macromolecular crowding at the ribosome exit site. For example, the abundant cotranslational chaperone trigger factor (TF) in bacteria can help the bacterial SRP reject borderline secretory protein substrates with weakly hydrophobic signal sequences [91]. Analogously to NAC, TF co-binds with SRP on the ribosome and regulates the activity of SRP via a multi-layered mechanism: it selectively reduces SRP–SR assembly rates on ribosomes displaying weakly hydrophobic signal sequences [91]. TF also restricts SRP-dependent targeting after the nascent polypeptide exceeds a critical length of ~130 amino acids, imposing a limited time window during translation for SRP to complete the targeting reaction [91]. This combination of allosteric and timing mechanisms allows TF to suppress the leaky cotranslational targeting of secretory proteins that can otherwise use the SecB/A post-translational translocation pathway. In another recent example, the specificity of an essential nascent protein modification enzyme in bacteria, methionine amino peptidase (MAP), was shown to be critically dependent on RPBs on the ribosome [92]. Cotranslational excision of the initiator methionine by MAP is rapid and diffusion-limited, with the irreversible chemical step significantly faster than the dissociation of MAP from the ribosome. As such, ribosome-bound MAP displays limited discrimination against suboptimal substrates with large side chains at the second amino acid [92]. A combination of RPBs, SRP and TF, selectively reduces the reaction rate of MAP at nascent chain lengths below 67 aa and beyond 82 aa [92]. This effectively restricts the action of MAP to a limited time window during translation elongation and thus re-establishes the sequence specificity of MAP during cotranslational processing of the nascent protein [92]. These and other recent work highlight the rich and dynamic mechanisms of molecular coordination between protein biogenesis factors on the ribosome and show that this coordination plays a vital role in ensuring the fidelity of nascent protein selection into their appropriate biogenesis pathways.

Recent findings in both co- and post-translational protein targeting pathways further emphasize the principle that the appropriate sorting of nascent proteins to cellular organelles is a result of the balanced action of multiple protein biogenesis factors and pathways. While mitochondrial proteins are mislocalized to the ER in the absence of NAC, acute depletion of SRP in yeast leads to the mistargeting of ribosomes translating normally ER-destined proteins to mitochondria, triggering rapid mitochondria fragmentation and dysfunction [7]. These results are reminiscent of the observations during the post-translational targeting of tail-anchored membrane proteins (TAs), in which deletion of components of the guided-entry-of-tail-anchored protein (GET) pathway resulted in the mistargeting of some ER-destined TAs to mitochondria [93]. These observations likely reflect the general tendency of hydrophobic, aggregation-prone membrane proteins to be mislocalized to membrane-enclosed organelles. They also suggest that any individual protein targeting pathway does not generate sufficient specificity of protein localization in the cell. Instead, organelle specificity of protein localization relies critically on the proper functioning of a combination of pathways and factors with opposing activities. These pathways likely possess overlapping yet distinct substrate preferences, which could enable more effective differentiation of degenerate targeting signals that share many physicochemical features.

While NAC provides a triage factor that facilitates the correct selection of translating ribosomes at early stages of ER targeting, it is likely that additional mechanisms are in place to ensure the fidelity of protein targeting and translocation to cellular membranes. In the case of bacterial SRP and MAP enzymes, kinetic rivalry with translation elongation proved to be an effective strategy to reject suboptimal substrates [91,92]; whether this principle operates in the mammalian SRP pathway to tune substrate selection remains to be determined. Intriguingly, the SRP9/14 subunits of the mammalian SRP competes with eEF1 and slows translation elongation (Figure 2A); whether and how this activity plays a role in the efficiency and substrate selection of SRP remain open questions [94,95,96,97,98,99,100,101]. In addition, early work showed that the Sec61p translocase provides a post-targeting mechanism to reject ribosomes with mutated signal sequences [102]. Subsequent biochemical and structural work revealed a lateral gate formed by TM2 and TM7 in the SecYEG/Sec61p complex that forms a docking site for TMDs and signal sequences [103,104], providing a molecular basis for the ability of this translocation machinery to recognize the targeting signal. Furthermore, surveillance and quality control pathways have been identified on both mitochondria and the ER that provide mechanisms for clearance of mislocalized membrane proteins. The conserved AAA-ATPase, Msp1 in yeast or ATAD1 in mammalian cells, localizes to the outer membrane of mitochondria and extracts ER-destined TAs that are mislocalized to mitochondria in the absence of a functioning GET pathway, as well as mislocalized peroxisomal TAs in the absence of the peroxisome targeting factor Pex19 [105,106,107,108,109]. Msp1 facilitates the transfer of mistargeted TAs from mitochondria to the ER, where the TA is recognized and degraded by the ubiquitin ligase Doa10 [110]. Reciprocally, an ER-resident P5A AAA-ATPase, ATP13A1 (Spf1 in yeast), recognizes and mediates the extraction of mitochondrial TAs mislocalized at the ER membrane [111,112]. While the observation of error-correction mechanisms has thus far used TAs as model substrates, whether analogous quality control machineries exist to correct mistakes in cotranslational protein targeting and to handle topologically more complex membrane proteins remain an outstanding question. The molecular mechanism by which these quality control machineries detect errors in protein localization also remain to be determined.

## Figures and Tables

**Figure 2 ijms-23-00281-f002:**
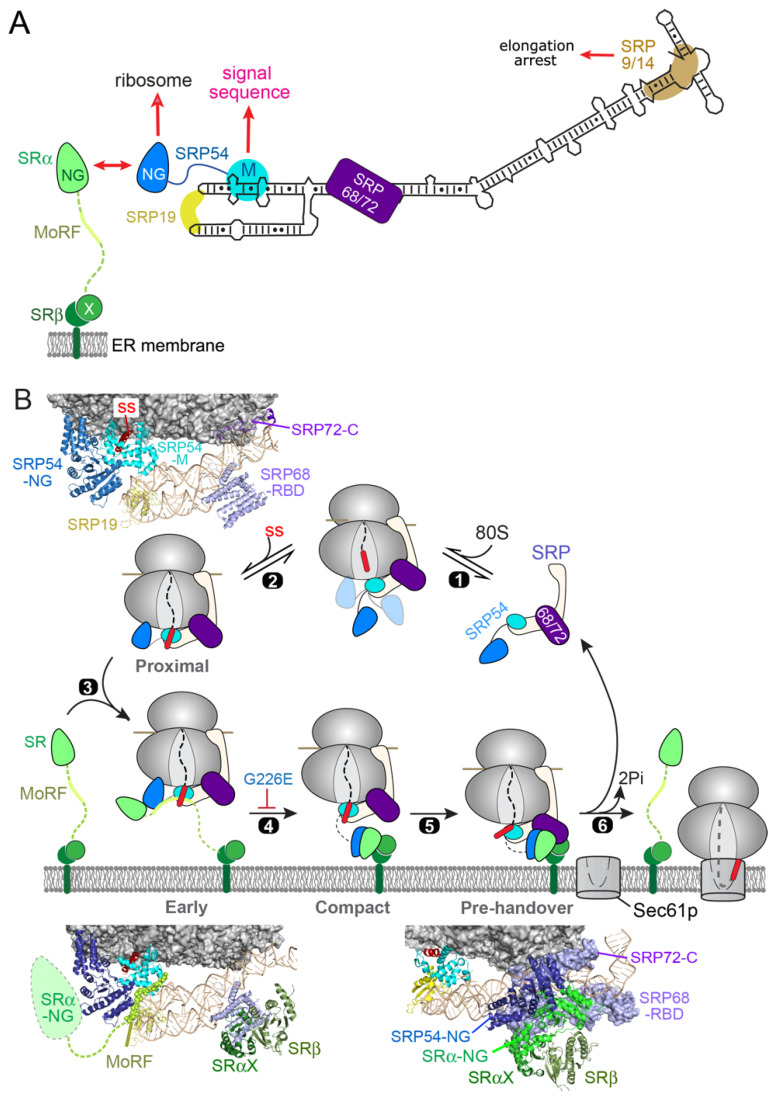
Model of the mammalian SRP pathway. (**A**) Schematic of the composition and interactions of the mammalian SRP and SRP receptor (SR). The individual subunits, domains, and important sequence motifs are defined in the text and indicated. MoRF, molecular recognition feature. (**B**) Current molecular model of the mammalian SRP pathway. Step 1, SRP binds to the translating ribosomes, on which it samples multiple conformations. Step 2, emergence of an ER signal sequence (*red*) drives SRP into the Proximal conformation. Step 3, early stage of SRP–SR assembly, mediated by dynamic interactions between the SRP and SR NG domains and by the SR MoRF (*lime*) interaction with SRP54. Step 4, a stable SRP/SR NG-heterodimer detaches from the ribosome exit site and docks onto the X/β domain of SR. Step 5, the NG•X/β complex docks onto the distal site of SRP to form the Pre-handover conformation, in which the translating ribosome is primed for handover to the Sec61p complex. Step 6, cargo is loaded on Sec61p to initiate protein translocation, and GTP hydrolysis drives the detachment of SRP from SR. The insets show the structural models of the RNC-SRP complex (PDB: 7OBR, upper left), the early RNC-SRP–SR complex (PDB: 7NFX, lower left), and RNC-SRP–SR pre-handover complex (PDB: 6FRK, lower right). Dashed outline in the early complex structure depicts SRα-NG that dynamically interacts with SRP54-NG at this stage and was not resolved in the structure. The M- and NG-domains of SRP54 are in *cyan* and *dark blue*, respectively; signal sequence (ss) is in red; SRP19 is in *yellow*, SRP68/72 is in *purple*, SRP RNA is in *tan*, SRαNG, SRαX, and SRβ are in *light green*, *dark green*, and *mustard*, respectively; MoRF in the SR linker is highlighted in spacefill model in *lime*.

**Figure 4 ijms-23-00281-f004:**
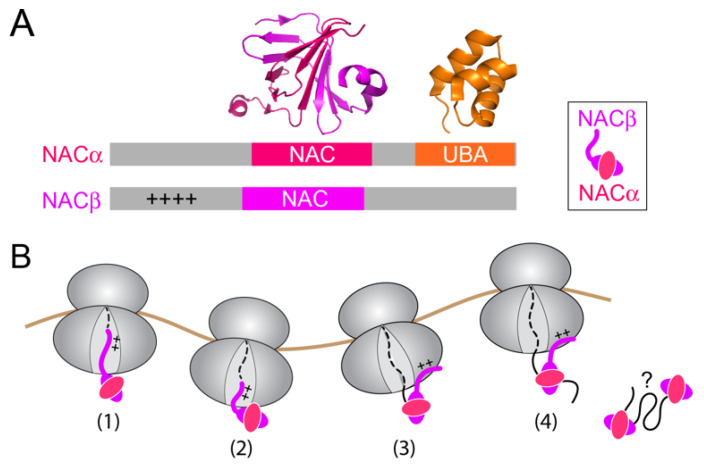
Proposed structure and interactions of NAC. (**A**) Overview of the domain composition and available structural information of NAC. Grey depicts unstructured extensions from the folded NAC and UBA domains. ‘++++’ denotes the basic RRKKK motif in the N-terminal extension of NACβ crucial for its ribosome binding. The crystal structures are shown for the C-terminal UBA domain of NACα (PDB: 1TR8) as well as the α/β NAC domain, which dimerizes into a β-barrel like structure (PDB: 3MCB). (**B**) Summary of models of NAC interaction with ribosome and the nascent chain during protein synthesis. NAC engages ribosome early during translation, with the N-terminal tail of NACβ inserting deeply into the exit tunnel of the ribosome (stage (1)). The inserted N-terminal tail of NACβ begins to retract in the tunnel as the nascent chain elongates to ~30 amino acids (stage (2)), and switches to interact with the ribosome surface upon the emergence of the nascent protein from the exit tunnel (stage (3)). NAC is likely anchored by the N-terminal NACβ tail on the ribosome, and the central NAC β-barrel could continue to contact portions of the nascent protein that just emerge from the tunnel exit as the nascent chain further elongates (stage (4)). Finally, NAC could also associate with aggregation prone proteins when released from the ribosome, although the molecular mechanism of NAC interactions off the ribosome awaits further investigation (?). ‘++++’ denotes the basic RRKKK motif at the N-terminus of NACβ.

**Figure 5 ijms-23-00281-f005:**
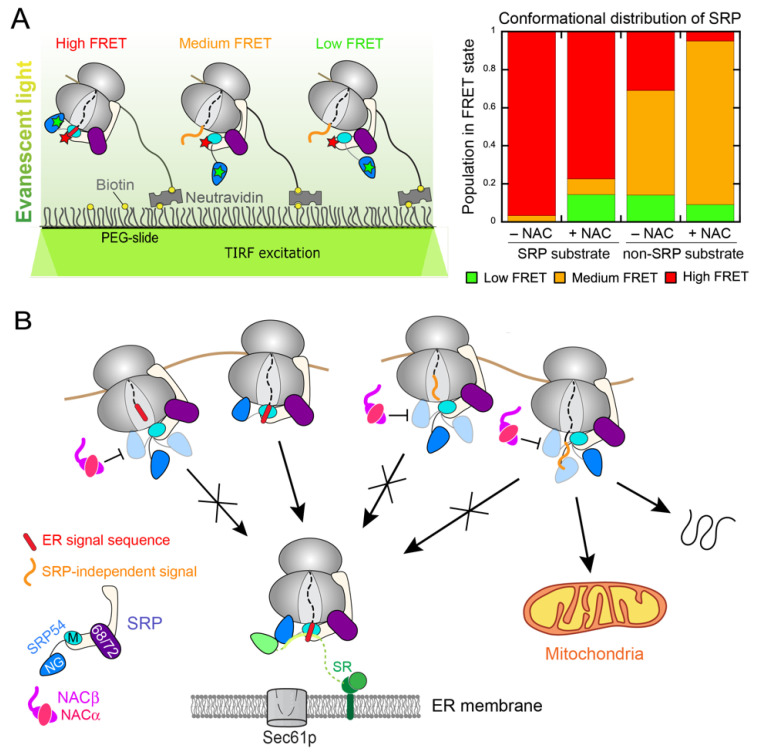
NAC enhances the specificity of ER targeting by regulating the conformation and activity of SRP. (**A**) Single-molecule FRET (smFRET) measurements show how NAC remodels the conformation of SRP on the ribosome. Left panel: schematic of the smFRET experiment. A FRET dye pair was incorporated on SRP54-NG and SRP19 to detect the Proximal conformation of SRP most active in SR recruitment. SRP was recruited to RNCs immobilized on the microscope slide surface, on which it can sample distinct conformational states that generate different FRET efficiencies between the dye pair. Right: summary of the conformational distribution of SRP on signal sequence-containing and signal-less ribosomes. NAC reduces the population of SRP in the high FRET state on signal-less ribosomes, directly demonstrating NAC allosteric regulation of SRP. Adapted from [56]. (**B**) Model of the mechanism by which NAC improves the targeting specificity of SRP. NAC (magenta) inhibits SRP from adopting the Proximal conformation on both short-chain RNCs and on ribosomes that expose a non-ER targeting signal, thus delaying the onset of ER targeting and preventing promiscuous targeting to the ER.

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
