# Peer review of "Fidelity of Cotranslational Protein Targeting to the Endoplasmic Reticulum"

_ijms, 2021, doi:10.3390/ijms23010281_

Round 1
Reviewer 1 Report
This is an excellent review on the fidelity of co-translational targeting to the endoplasmic reticulum (ER). I appreciate how the Introduction lays out the problems/challenges of co-translational targeting to the ER, the perspectives provided at the end of the review, and the overall clarity of the manuscript. I only have minor comments, mainly related to typographical issues.
MINOR ISSUES
- In the legend to Figure 1, "hydrophobic sequences" is abbreviated "SS". Perhaps "hydrophobic signal sequences" would be a better phrase for the abbreviation.
- The "eL" and "uL" designations in Figure 1B should be defined in the legend.
- on p. 3, the paragraph from lines 79-89 is in a different font (and perhaps size as well) than the rest of the text.
- p. 5 line 139: remove the hyphen in "mole-cular".
- The last sentence of the Figure 3 legend on p. 6 is repeated as the first sentence on p. 7, where the legend continues.
- If the data presented in Figure 3 are published, this should be noted in the Figure legend with a reference. If it has not been published, more experimental detail on the protocol should be provided. This comment also applies to Figure 5A.
- In the Figure 4 legend, NACβ is twice written as NACb (lines 241 and 243).
- The RNC abbreviation is used on p. 7 line 225, but RNC is not defined until p. 8 line 281.
- In the Figure 4 legend, the phrase "…, with the N-terminal tail of NACβ inserts deeply into the exit tunnel" is awkward. Perhaps switch to either "…, when the N-terminal tail of NACβ inserts deeply into the exit tunnel" or "…, with the N-terminal tail of NACβ inserting deeply into the exit tunnel".
- In the Figure 4 legend, does the phrase "could continue to contact portions of the nascent protein" (p. 8 line 244) refer to contact between the dimerized β-barrel structure of NAC and the nascent protein? This seems to be the implication from the image in Figure 4B, but it is not explicitly stated in the legend.
- On p. 8 lines 248-250, the text is a different shade of black than the rest of the text.
Author Response
Reviewer 1
This is an excellent review on the fidelity of co-translational targeting to the endoplasmic reticulum (ER). I appreciate how the Introduction lays out the problems/challenges of co-translational targeting to the ER, the perspectives provided at the end of the review, and the overall clarity of the manuscript. I only have minor comments, mainly related to typographical issues.
MINOR ISSUES
- In the legend to Figure 1, "hydrophobic sequences" is abbreviated "SS". Perhaps "hydrophobic signal sequences" would be a better phrase for the abbreviation.
Corrected.
- The "eL" and "uL" designations in Figure 1B should be defined in the legend.
Defined.
- on p. 3, the paragraph from lines 79-89 is in a different font (and perhaps size as well) than the rest of the text.
Corrected.
- p. 5 line 139: remove the hyphen in "mole-cular".
Corrected.
- The last sentence of the Figure 3 legend on p. 6 is repeated as the first sentence on p. 7, where the legend continues.
Corrected.
- If the data presented in Figure 3 are published, this should be noted in the Figure legend with a reference. If it has not been published, more experimental detail on the protocol should be provided. This comment also applies to Figure 5A.
We have added the references to these figure panels in the figure legend.
- In the Figure 4 legend, NACβ is twice written as NACb (lines 241 and 243).
Corrected.
- The RNC abbreviation is used on p. 7 line 225, but RNC is not defined until p. 8 line 281.
We moved the RNC definition to the first time the term was used.
- In the Figure 4 legend, the phrase "…, with the N-terminal tail of NACβ inserts deeply into the exit tunnel" is awkward. Perhaps switch to either "…, when the N-terminal tail of NACβ inserts deeply into the exit tunnel" or "…, with the N-terminal tail of NACβ inserting deeply into the exit tunnel".
We revised the text following the reviewer’s suggestion (‘… inserting deeply…’)
- In the Figure 4 legend, does the phrase "could continue to contact portions of the nascent protein" (p. 8 line 244) refer to contact between the dimerized β-barrel structure of NAC and the nascent protein? This seems to be the implication from the image in Figure 4B, but it is not explicitly stated in the legend.
We clarified this sentence, which indeed refers to the NAC b-barrel domain.
- On p. 8 lines 248-250, the text is a different shade of black than the rest of the text.
Corrected.
Reviewer 2.
This review describes the molecular mechanisms that regulated fidelity of protein targeting to the endoplasmic reticulum during translation. It provides very detailed overview, is very informative and is quite well written. However, while this is clearly a review article, it seems to contain some original experimental data (in the form of figures derived from data previously published by the same authors – ie Fig 3 and Fig 5A), for which the methodology is not detailed here. I am not sure if the journal accepts mixed review and research articles. My comments are generally quite minor, and I think the manuscript could benefit by improving the figures as described in my detailed comments.
This is the same comment as point 6 by reviewer 1. We have added the references for the original data in the figure legend, as suggested by the reviewer.
Line 47: Regarding the statement ‘this time window is significantly longer in eukaryotic cells and could increase the probability of mis-targeting’ This does not necessarily follow, and therefore this statement could be misleading, since a longer time window could also decrease the probability of mis-targeting. The authors need to amend this statement.
I am not sure what is missing here. This sentence is preceded by “translation termination or a critical length of the nascent protein imposes a time window for SRP to complete the targeting reaction”. Optimal substrates are targeted rapidly and insensitive to variations in the time window. Suboptimal substrates, which is targeted more slowly, will be more effectively rejected if the time window is short, whereas a longer time window will lead to increased chances for the suboptimal substrates to leak through the system. We explained the concept of the time window in more detail in the text, and hope it is clearer now.
Fig 1B: it is difficult to see the different overlays and the colour choices are perhaps not ideal particularly if someone is colour blind. I cannot confidently identify MetAP2. I suggest that each overlay is labelled with an arrow and its name. What are the eL numbers on the figure? ‘The contours of ribosomal proteins surrounding the exit site are indicated.’ To me it is not clear where these contours are.
We changed the coloring of MetAP2 and hope that it is easier to see. Our initial version of Figure 1B had each RPB labeled on the figure, as suggested by the reviewer. However, because of the large number of factors and ribosomal proteins that need to be indicated, this just made a messy figure that is much harder to see than the current version.
Line 95: Can the authors clarify what is intended by ‘SRP RNA’?
SRP is a ribonucleoprotein complex and contains an SRP RNA. In eukaryotes the RNA serves as a scaffold on which the SRP protein subunits bind (see Figure 2). We clarified this in the text.
Fig 2A and legend: the authors need to define SR here, since it is the first mention after Fig 2A is referenced in line 96. Also, other terms that are in Fig 2A need to be defined in the legend, including MoRF, SRbeta, SRalpha, SRP9/14, S-domain, Alu-domain. Here and in other figures, the words MoRF and SRP19 are very difficult to read because of the way they are presented as outline text. Please change to something easier for the reader to see.
We defined SR and MoRF in the figure 2 legend. Other terms are defined in the text and are self-obvious. For example, SRP14 = the SRP14 subunit of SRP. SRa = the a subunit of SR.
We removed the S-domain and Alu-domain labels in Fig. 2A, as the concepts are not necessary for the discussion here.
We changed the color of the MoRF and SRP19 labels.
Line 105: it looks like there is something missing from ‘signal sequence not only bind SRP more strongly’. Can the authors please correct?
There is no problem with this sentence. The complete sentence is copied below.
“ribosomes bearing an SRP-dependent signal sequence not only bind SRP more strongly, but also mediate SRP-SR assembly at rates that are 100-1000 fold faster than those on signalless ribosomes or ribosomes with suboptimal signal sequences”.
Fig 2B: What/where is the ‘X/b domain of SR’ mentioned in the fig legend? The color scheme is difficult to follow – I can see dark green, medium green (what is that?), light green, lime (light and dark), but I cannot be sure which is for which part of the complex. I also see light and dark purple on the figure – only purple is mentioned in the legend. Can this be made more clear? Perhaps a color legend on the figure would help with matching the different shades to the proteins etc.
These terms were defined in the text: “eukaryotic SR is a heterodimer of SRα and SRβ subunits (Fig. 2A). SRβ is a single-pass transmembrane protein anchored at the ER. SRα binds tightly to SRβ via its N-terminal X-domain[47, 48], which is connected to the NG-domain through a ~200-residue intrinsically disordered linker that contains sites for ribosome interaction and sensing[49, 50].”
Line 129 - mole- cular. Please correct
Corrected.
Line 176: what is the symbol (xx) after ‘pre-handover complex (Fig. 2B, xx)’? ‘
The inhibition sign in Figure 2B, underneath ‘G226E’.
Line 203: repetition of NAC enhances the specificity of human SRP during the SR recruitment step from < 2-fold to ~50- 203 fold.
Corrected.
The figures and figure parts should be presented in the sequence that they are referred to in the text (eg fig 4b should be presented after 4a, fig 4 should be presented after all parts of fig 3 have been referred to)
It is natural that different parts of a figure (which naturally belong together in terms of the nature of the information being presented) will be presented at different points of a discussion. In this context, panels 3C and 3D (with NAC present) are better grouped with those in 3A and 3B (without NAC present) because the nature of the data presented is the same or similar (measurements of targeting kinetics). It is easiest to see the comparison when the data are together in the same figure, even though the discussion of the effects of NAC came later in the text.
Line 220: there is no ++ in Fig 4A, rather there is ++++ To avoid confusion can the authors be more precise?
Done.
Line 241 and 243: should NACb be NAC beta?
Corrected to NACb.
Line 256: should pucta be puncta?
Corrected.
General - there are different fonts in different parts of the manuscript
Corrected.
Some of the text on some of the figures is very difficult to read, in particular those where the words are formatted in outline (eg SRP19 on Fig 2A). These should be changed.
Corrected.
Reviewer 2 Report
This review describes the molecular mechanisms that regulated fidelity of protein targeting to the endoplasmic reticulum during translation. It provides very detailed overview, is very informative and is quite well written. However, while this is clearly a review article, it seems to contain some original experimental data (in the form of figures derived from data previously published by the same authors – ie Fig 3 and Fig 5A), for which the methodology is not detailed here. I am not sure if the journal accepts mixed review and research articles. My comments are generally quite minor, and I think the manuscript could benefit by improving the figures as described in my detailed comments.
Line 47: Regarding the statement ‘this time window is significantly longer in eukaryotic cells and could increase the probability of mis-targeting’ This does not necessarily follow, and therefore this statement could be misleading, since a longer time window could also decrease the probability of mis-targeting. The authors need to amend this statement.
Fig 1B: it is difficult to see the different overlays and the colour choices are perhaps not ideal particularly if someone is colour blind. I cannot confidently identify MetAP2. I suggest that each overlay is labelled with an arrow and its name. What are the eL numbers on the figure? ‘The contours of ribosomal proteins surrounding the exit site are indicated.’ To me it is not clear where these contours are.
Line 95: Can the authors clarify what is intended by ‘SRP RNA’?
Fig 2A and legend: the authors need to define SR here, since it is the first mention after Fig 2A is referenced in line 96. Also, other terms that are in Fig 2A need to be defined in the legend, including MoRF, SRbeta, SRalpha, SRP9/14, S-domain, Alu-domain. Here and in other figures, the words MoRF and SRP19 are very difficult to read because of the way they are presented as outline text. Please change to something easier for the reader to see.
Line 105: it looks like there is something missing from ‘signal sequence not only bind SRP more strongly’. Can the authors please correct?
Fig 2B: What/where is the ‘X/b domain of SR’ mentioned in the fig legend? The color scheme is difficult to follow – I can see dark green, medium green (what is that?), light green, lime (light and dark), but I cannot be sure which is for which part of the complex. I also see light and dark purple on the figure – only purple is mentioned in the legend. Can this be made more clear? Perhaps a color legend on the figure would help with matching the different shades to the proteins etc.
Line 129 - mole- cular. Please correct
Line 176: what is the symbol (xx) after ‘pre-handover complex (Fig. 2B, xx)’? ‘
Line 203: repetition of NAC enhances the specificity of human SRP during the SR recruitment step from < 2-fold to ~50- 203 fold.
The figures and figure parts should be presented in the sequence that they are referred to in the text (eg fig 4b should be presented after 4a, fig 4 should be presented after all parts of fig 3 have been referred to)
Line 220: there is no ++ in Fig 4A, rather there is ++++ To avoid confusion can the authors be more precise?
Line 241 and 243: should NACb be NAC beta?
Line 256: should pucta be puncta?
General - there are different fonts in different parts of the manuscript
Some of the text on some of the figures is very difficult to read, in particular those where the words are formatted in outline (eg SRP19 on Fig 2A). These should be changed.
Author Response

(The authors gave the same response as above.)
